analytical chemistry/medicinal chemistry/spectroscopy

nazartinib, EGF816, human serum albumin, fluorescence quenching

**Author for correspondence:**
Ali Saber Abdelhameed
e-mail: asaber@ksu.edu.sa

# Spectroscopic and molecular docking studies reveal binding characteristics of nazartinib (EGF816) to human serum albumin

Abdulrahman A. Almehizia[1], Haitham AlRabiah[1], Ahmed H. Bakheit[1,2], Eman S. G. Hassan[3], Rashed N. Herqash[1] and Ali Saber Abdelhameed[1]

[1]Department of Pharmaceutical Chemistry, College of Pharmacy, King Saud University, PO Box 2457, Riyadh 11451, Saudi Arabia
[2]Department of Chemistry, Faculty of Science and Technology, El-Neelain University, PO Box 12702, Khartoum 11121, Sudan
[3]Developmental Pharmacology Department, National Organization for Drug Control and Research, Giza, Egypt

AAA, 0000-0001-8711-3873; ESGH, 0000-0003-0655-1869; ASA, 0000-0002-5910-2832

The interactions of novel anti-cancer therapeutic agents with the different plasma and tissue components, specifically serum albumins, have lately gained considerable attention due to the significant influence of such interactions on the pharmacokinetics and/or -dynamics of this important class of therapeutics. Nazartinib (EGF 816; NAZ) is a new anti-cancer candidate proposed as a third-generation epidermal growth factor receptor tyrosine kinase inhibitor that is being developed and clinically tested for the management of non-small cell lung cancer. The current study aimed to characterize the interaction between NAZ and human serum albumin (HSA) using experimental and theoretical approaches. Experimental results of fluorescence quenching of HSA induced by NAZ revealed the development of a statically formed complex between NAZ and HSA. Interpretation of the observed fluorescence data using Stern–Volmer, Lineweaver–Burk and double-log formulae resulted in binding constants for HSA-NAZ complex in the range of $(2.34–2.81) \times 10^4 \, M^{-1}$ over the studied temperatures. These computed values were further used to elucidate thermodynamic attributes of the interaction, which showed that NAZ spontaneously binds to HSA with a postulated electrostatic force-driven interaction. This was further verified by theoretical examination of the NAZ

docking on the HSA surface that revealed an HSA-NAZ complex where NAZ is bound to HSA Sudlow site I driven by hydrogen bonding in addition to electrostatic forces in the form of pi-H bond. The HSA binding pocket for NAZ was shown to encompass ARG 257, ARG 222, LYS 199 and GLU 292 with a total binding energy of $-25.59$ kJ mol$^{-1}$.

# 1. Introduction

The last decade has witnessed an evolution in the number of molecularly targeted therapies, particularly those molecules aimed at the different types of tumours. Scientific progress in understanding the nature and genesis of these diseases has led to improved cancer survival rates. A significant number of those molecularly targeted entities has been officially permitted by the different regulatory authorities for treatment of several types of cancer either as small molecule drugs or monoclonal antibodies [1,2]. In the same context, 37 targeted kinase inhibitors have been granted FDA approval for various types of cancer such as kidney, breast, colon and liver cancers, with an additional 150 kinase-targeted agents being clinically tested, and numerous moieties in preclinical assessment [2]. A new member of the family of kinase inhibitors is nazartinib (figure 1; NAZ; EGF816), a third-generation epidermal growth factor receptor (EGFR) tyrosine kinase inhibitor (TKI) developed by Novartis Oncology that is being clinically tested for the treatment of solid malignancies, with a focus on non-small cell lung cancer (NSCLC) [4–6]. Preclinical observations demonstrated that NAZ possesses mutant-specificity and wild-type sparing characteristics against EGFR similar to former third-generation EGFR TKIs [4]. NAZ is currently being examined in combination with various drugs against several malignancies with a focus on EGFR-mutant NSCLC such as in phase I/II study with the c-MET inhibitor INC280 (capmatinib) (NCT02335944), and in a phase II study with trametinib against NSCLC with T790M-positive resistance to EGFR TKI therapy (NCT03516214) [7].

Detailed investigation of the interactions of important classes of therapeutics as well as of other potential drugs with either plasma or target tissue proteins, particularly with serum albumins, has been considered an important part of pharmacological profiling [8–11]. Human serum albumin (HSA) is the most abundant and most studied carrier protein in the plasma. HSA binds a wide array of exogenous and endogenous chemicals, facilitating their conveyance throughout the body. Therefore, the current study was aimed at understanding the interaction between NAZ and HSA and at elucidating molecular mechanisms underlying this interaction. This study employed fluorescence and UV–visible (UV–vis) spectral observations, and theoretical docking studies to better understand the HSA-NAZ interaction.

# 2. Material and methods

## 2.1. Standards and reagents

Unless otherwise stated, all chemicals, reagents, buffer constituents and solvent were procured from Sigma-Aldrich Co. (St Louis, MO, USA). Nazartinib (NAZ) standard powder with a 98.03% purity was obtained from MedChemExpress (Princeton, NJ, USA). Double-distilled and de-ionized water from a Milli-Q® UF-Plus purification system (Millipore, Bedford, MA, USA) was used throughout the study.

## 2.2. Preparation of experimental solutions

Due to its sparingly soluble characteristic in aqueous solvents, NAZ was dissolved in ethanol at a concentration of 25 mM as stock solution. This solution was diluted afterwards in 1X PBS (phosphate-buffered saline), pH 7.4, to produce the various NAZ working solutions. Fatty acid free HSA solution was prepared in the same buffer at a concentration of 1.5 µM with the protein content verified by a Schimadzu™ UV-1800 spectrophotometer (Schimadzu Co., Tokyo, Japan).

## 2.3. Fluorescence-based measurements

Three different experimental settings based on the native fluorescence of HSA were employed during the present study, namely, emission, synchronous and three-dimensional fluorescence measurements. All

**Figure 1.** Two-dimensional chemical structure of NAZ in its theoretically postulated ionized form at pH 7.4, calculated from chemcalize.com [3].

observations were executed using a 1 cm quartz cuvette and a Jasco FP-8200 (Jasco Int. Co. Ltd, Tokyo, Japan) spectrofluorometer.

After excitation of the HSA molecules at $\lambda_{ex}$ 280 nm, the intensity of the emitted light was recorded over a wavelength range of 290–500 nm, with excitation and emission slit widths set at 5 nm. The experimental procedure for emission and synchronous fluorescence assays were executed at 298, 303 and 310 K for 1.5 µM HSA solutions mixed with NAZ solutions of 0, 1.8, 3.6, 5.5, 7.4, 15.0 and 22.0 µM. Synchronous fluorescence assays were performed at two different interval wavelengths ($\Delta\lambda$), namely 15 nm and 60 nm, which signify the characteristics of the micro-surroundings of tyrosine (Tyr) and tryptophan (Trp), respectively, in the protein structure. Three-dimensional fluorescence observations were executed with a 1.5 µM solution of HSA in combination with 0 or 5.5 µM of NAZ in solution at 298 K. For such measurements, excitation and emission ranges were set to 210–340 nm for $\lambda_{ex}$ and 240–600 nm for $\lambda_{em}$ scanning every 2 nm. Furthermore, to distinguish the emission response of the samples from the response of light-absorbing artefacts in the assay (known as the inner filter effect), $\lambda_{ex}$ and $\lambda_{em}$ (known as the inner filter effect), all recorded fluorescence responses ($F_{obs}$) were adjusted ($F_{cor}$) using the UV–vis absorbance values ($A_{ex}$ and $A_{em}$) of NAZ at the assay $\lambda_{ex}$ and $\lambda_{em}$, respectively, using equation (2.1) [12,13].

$$F_{cor} = F_{obs} \times e^{(A_{ex}+A_{em})/2}. \tag{2.1}$$

Emission measurements were additionally evaluated to examine the specific binding site of NAZ within the HSA structure in the proposed HSA-NAZ complex by the inclusion of previously proven HSA site markers, namely phenylbutazone (PHB) and ibuprofen (IBP), marking sites I and II of HSA, respectively. For these experiments, solutions of 1.5 µM HSA and NAZ (0, 1.8, 3.6, 5.5, 7.4, 15.0 and 22.0 µM) were used with the addition of 1.5 µM of either PHB or IBP prior to the NAZ addition.

## 2.4. UV–vis spectral observations

Various reports have shown the utility of UV–vis absorption measurements for drug-protein interaction assays [14–17]. This approach was mainly aimed at further confirming the statically formed protein-drug complex and observing protein structural changes once it is bound to the ligand. Therefore, the proposed complex of HSA-NAZ was assessed using UV–vis spectrophotometry (Schimadzu™ UV-1800, Schimadzu Co., Tokyo, Japan). The spectra were logged over a wavelength of 200–500 nm for separate solutions of 1.5 µM HSA and 3.7 µM NAZ, as well as solutions of HSA (1.5 µM) with varying concentrations (3.7, 5.5 or 7.4 µM) of NAZ.

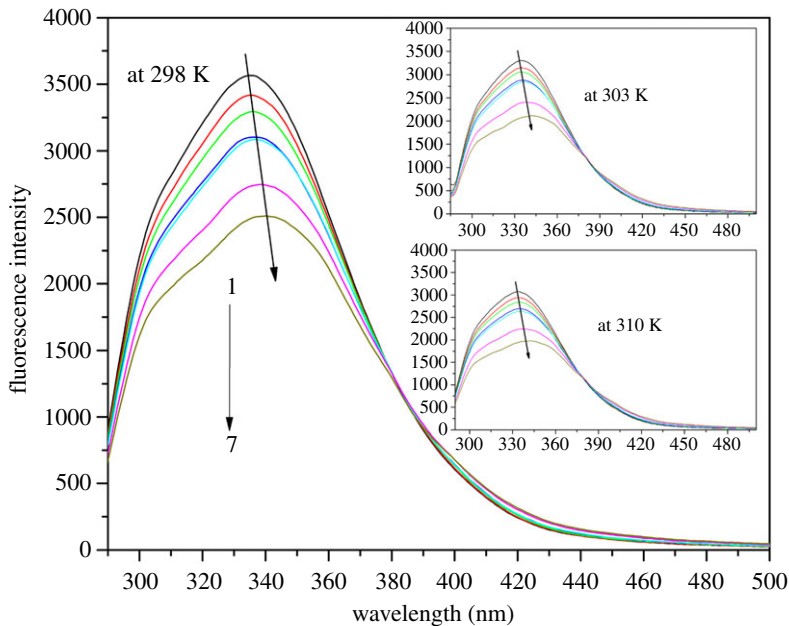

**Figure 2.** The recorded fluorescence response of HSA with various concentrations of NAZ (at 0, 1.8, 3.6, 5.5, 7.4, 15.0 and 22.0 μM as curves 1–7, respectively).

## 2.5. Molecular docking

To complement the experimental results, computer-based docking assessment of the HSA and NAZ binding was executed. This study was performed using previously reported crystal structure of HSA complexed with PHB (PDB ID: 2BXD) [18] and IBP (PDB ID: 2BXG) [18] with a three-dimensional NAZ structure produced by ChemDraw® Ultra 14.0 (Perkin Elmer Informatics, MA, USA). Prior to the identification of the docked poses of NAZ, the examined HSA and NAZ structures were adjusted by energy minimization of NAZ and elimination of heteroatoms and water molecules with the addition of hydrogen atoms to HSA. Structural prior-optimization and docking investigations were performed using the Molecular Operating Environment software package (MOE® 2014, Chemical Computing Group ULC, QC, Canada).

# 3. Results and discussion

## 3.1. HSA fluorescence quenching

A concentration-dependent quenching of the intrinsic fluorescence of HSA by NAZ was detected upon spectral monitoring of the fluorescence intensity of the HSA-NAZ interaction (figure 2). Significant quenching of a protein's fluorescence intensity following its interaction with a ligand is typically due to either static formation of a non-fluorescent complex or to dynamic molecular diffusion.

Hence, in order to differentiate between the two mechanisms, the recorded spectral results were analysed with the aid of three well-known formulae, namely the Stern–Volmer, Lineweaver–Burk and the double-log relations (equations (3.1), (3.2) and (3.4)) [19–21]. Since the two mechanisms can be distinguished based on the relationship between the binding constants and temperature, spectral data of the HSA-NAZ interaction was observed at different temperatures (298, 303 and 310 K). Generally, this escalation in temperature would result in higher binding constant values for the dynamic type of quenching, while a decrease in the binding constant would indicate a static complex formation [13,22,23].

$$\frac{F_0}{F} = 1 + K_{\mathrm{SV}}C_Q = 1 + k_q\tau_0 C_Q, \tag{3.1}$$

$$(F_0 - F)^{-1} = F_0^{-1} + K_{\mathrm{LB}}^{-1}F_0^{-1}C_Q^{-1}, \tag{3.2}$$

$$k_q = \frac{K_{\mathrm{SV}}}{\tau_0} \tag{3.3}$$

and

$$\log\left(\frac{F_0 - F}{F}\right) = \log K + n\log C_Q. \tag{3.4}$$

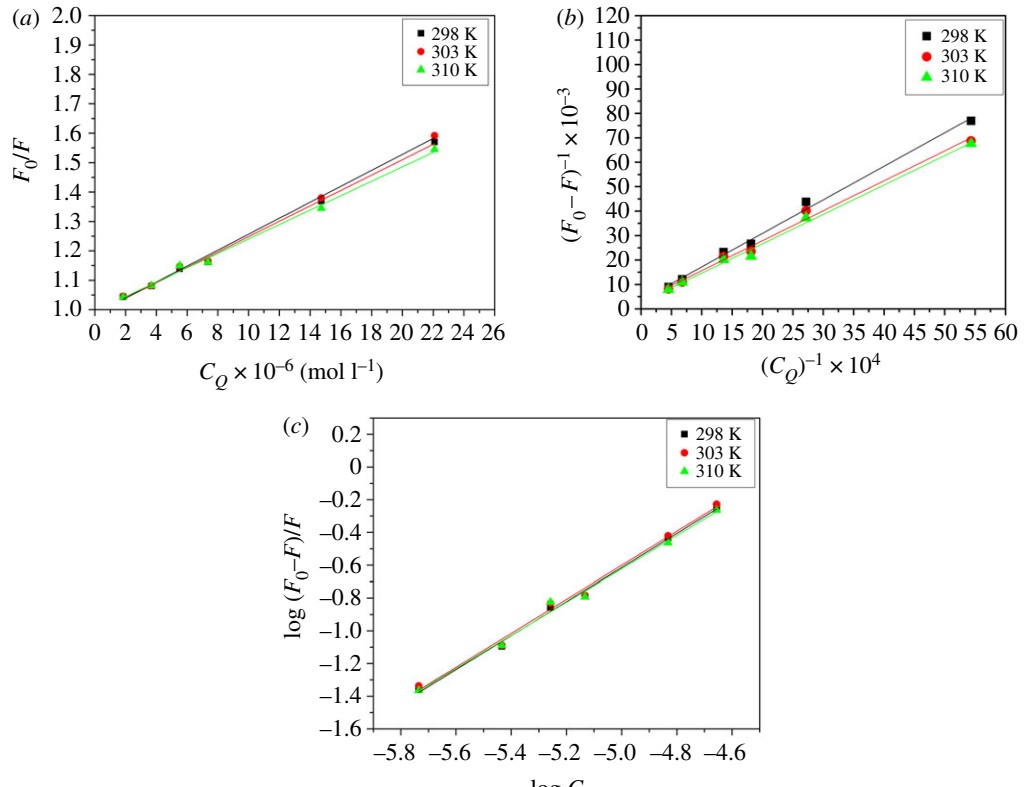

**Figure 3.** (a) Stern–Volmer, (b) Lineweaver–Burk and (c) the double-log graphs of the experimental fluorescence data for HSA-NAZ binding.

In these equations (equations (3.1)–(3.4)), values referring to the spectral intensity of HSA-only and HSA-NAZ complex are abbreviated to $F_0$ and $F$, respectively. The concentration of NAZ is referred to as $C_Q$ and the Stern–Volmer, Lineweaver–Burk and double-log equation constants are signified as $K_{SV}$, $K_{LB}$ and $K$, respectively, with the number of binding sites represented as $n$. The quenching rate constant is symbolized by $k_q$, while $\tau_0$ represents the mean protein lifetime with no quencher and considered as approximately 2.7 ns [24]. The spectral results demonstrated linear fitting for the data points in all plots derived from equations (3.1), (3.2) and (3.4) (figure 3a–c) which, in turn, primarily indicate a static type of quenching induced by the interaction of NAZ with HSA. This observation was supported by the temperature-dependent decrease in the computed constants based on the three plots (table 1). Similarly, the $k_q$ values determined through equation (3.3) ranged from $0.91 \times 10^{13}$ to $1.00 \times 10^{13}\,\mathrm{M^{-1}\,s^{-1}}$ (table 1), which is greater than the previously reported value of $2 \times 10^{10}\,\mathrm{M^{-1}\,s^{-1}}$ [25] for a dynamic quenching process of a macromolecule. This, thereby, further supports the steady complex formation mechanism between HSA and NAZ [26].

## 3.2. Thermodynamic features of NAZ-HSA interaction

The characteristics of the static complex formation were investigated to further understand the mechanism(s) involved in the HSA-NAZ interaction. Hence, estimation of the changes in the major thermodynamic attributes, namely the Gibbs free energy ($\Delta G°$), enthalpy ($\Delta H°$) and entropy ($\Delta S°$) changes was performed. Equations (3.5) and (3.6) were employed to compute these thermodynamic values using the gas constant $R$, the experimental temperature $T$ and the calculated $K$ values in table 1. Subsequent plotting of ln K ($y$-axis) against $1/T$ ($x$-axis) (figure 4), produced a linear fit of the data points. Then, the values of $\Delta G°$, $\Delta H°$ and $\Delta S°$ were determined from the resulting linear regression equation as enumerated in table 2. Several earlier studies have used these thermodynamic attributes to discover the potential binding forces between various ligands and proteins [24,27,28]. These reports claim that the sign and magnitude of individual and/or combined values of entropy and enthalpy strongly indicate the type of non-covalent binding forces behind molecular interactions as shown in the diagrammatic illustration (figure 5). Consequently, the observed results in the current

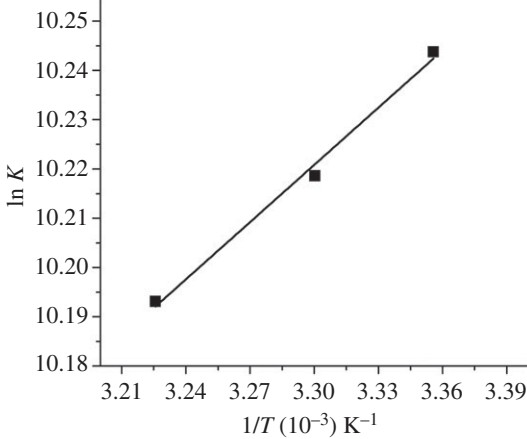

**Figure 4.** Van't Hoff graph for HSA-NAZ binding.

**Table 1.** Summarized binding parameters for NAZ and HSA. Values are mean of three independent experiments.

| | Stern–Volmer parameters | | | Lineweaver–Burk parameters | | binding parameters | | |
|---|---|---|---|---|---|---|---|---|
| $T$ (K) | $K_{SV} \times 10^4$ (M$^{-1}$) | $k_q \times 10^{13}$ (M$^{-1}$ s$^{-1}$) | $r^2$ | $K_{LB} \times 10^4$ (M$^{-1}$) | $r^2$ | $K \times 10^4$ (M$^{-1}$) | $n$ | $r^2$ |
| 298 | $2.71 \pm 0.08$ | 1.00 | 0.9982 | $2.79 \pm 0.10$ | 0.9959 | $2.81 \pm 0.07$ | $1.04 \pm 0.03$ | 0.9979 |
| 303 | $2.62 \pm 0.06$ | 0.97 | 0.9989 | $2.54 \pm 0.07$ | 0.9976 | $2.74 \pm 0.06$ | $1.04 \pm 0.04$ | 0.9965 |
| 310 | $2.45 \pm 0.08$ | 0.91 | 0.9987 | $2.34 \pm 0.06$ | 0.9971 | $2.67 \pm 0.08$ | $1.01 \pm 0.04$ | 0.9963 |

study (table 2) advocate that an electrostatic-driven interaction spontaneously takes place between NAZ and HSA, yet further docking studies can provide more detailed information on the proposed interaction forces between these two molecules.

$$\Delta G^\circ = -RT\ln K = \Delta H^\circ - T.\Delta S^\circ \tag{3.5}$$

and

$$\ln K = -\frac{\Delta H^\circ}{RT} + \frac{\Delta S^\circ}{R}. \tag{3.6}$$

## 3.3. Synchronous and three-dimensional spectral studies

The pioneer work reported by Lloyd in the early 1970s [29] demonstrated the use of the synchronous fluorescence approach that is now considered a standard way to monitor the influence of various ligands on the inherent fluorescence of a protein, particularly with regard to polarity alterations in the binding pocket. Previous studies have determined that measurements performed at $\Delta\lambda$ values of 15 and 60 nm for a given protein are linked to modifications in polarity near Tyr or Trp, respectively [30,31]. Therefore, shifts in the synchronous peaks at any of these $\Delta\lambda$ values indicate changes in the microenvironment of the corresponding residues. That is to say, a peak shift to a longer wavelength (bathochromic) or shorter wavelength (hypsochromic) would suggest reduced or increased hydrophobicity, respectively, around Tyr and Trp [32,33]. In the present study, although the fluorescence emission spectra in figure 2 showed peak shift, no shifts were observed in the synchronous spectra for the HSA-NAZ interaction, which may revert the shift in the total fluorescence emission spectra to intrinsic fluorescence of the ligand at higher wavelength values. A steady decline in the peak intensity was noted at both $\Delta\lambda$ values (figure 6), signifying unchanged surroundings for both Tyr and Trp. The results obtained from three-dimensional measurements also showed that the binding of NAZ to HSA led to a reduced intensity of the inherent fluorescence of HSA compared to the native protein (figure 7). Additionally, two defined three-

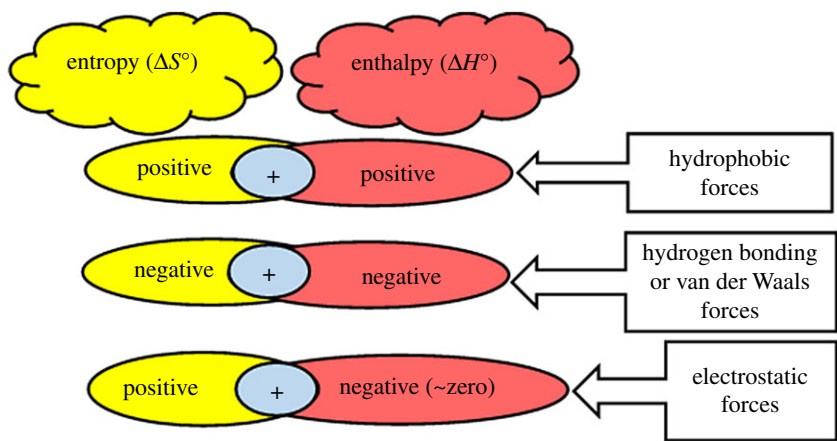

**Figure 5.** Diagrammatic representation of the postulated interaction forces based on thermodynamic parameters.

**Table 2.** Thermodynamic parameters for HSA-NAZ as calculated from fluorescence data. Values are mean of three independent experiments.

| $T$ (K) | $\Delta G°$ (kJ mol$^{-1}$) | $\Delta H°$ (kJ mol$^{-1}$) | $\Delta S°$ (J mol$^{-1}$ K$^{-1}$) |
|---|---|---|---|
| 298 | $-25.37 \pm 0.82$ | $-3.22 \pm 0.12$ | $74.36 \pm 3.24$ |
| 303 | $-25.75 \pm 0.60$ | | |
| 310 | $-26.27 \pm 0.76$ | | |

dimensional fluorescence peaks observed in the HSA native fluorescence at $\lambda_{\mathrm{ex}}/\lambda_{\mathrm{em}}$ 234/336 nm and at 280/336 nm were subsequently quenched upon NAZ binding (table 3). The two peaks were previously designated as signifying the $n \to \pi^*$ conversion of the polypeptide backbone (peak 1 at 234/336 nm) and of the Trp and Tyr residues (peak 2 at 280/336 nm) [34–36].

## 3.4. UV–vis spectral observations

The UV–vis spectra of the HSA-NAZ complex as well as those of NAZ and HSA individually were also monitored. Changes in the HSA peak intensity and shape upon binding of HSA to NAZ in the NAZ-subtracted HSA spectrum in figure 8 provide additional evidence for the HSA-NAZ complex formation. These conformational changes and the concentration-dependent increase in the UV–vis response of the HSA-NAZ complex are consistent with the fluorescence-based results that support the static binding between NAZ and HSA.

## 3.5. Markers of the binding sites

Taken together, the results of this study confirm that a static binding takes place between NAZ and HSA in solution. To identify the binding site of NAZ on the HSA surface, we examined the ability of NAZ to displace markers of HSA Sudlow sites I and II [37], namely phenylbutazone (PHB) and ibuprofen (IBP), respectively [18]. Analysis of the obtained HSA-NAZ fluorescence spectra in the presence and absence of IBP and PHB, using the Stern–Volmer equation (equation (3.1)) and its derived double-log equation (equation (3.4)), was performed, and data were plotted accordingly (figure 9). The computed $K_{\mathrm{sv}}$ and $K$ values in table 4 show that NAZ competes with PHB for the HSA Sudlow site I, while there was no alteration in the IBP binding affinity to HSA. Therefore, these results suggest that NAZ binds to the Sudlow site I on the HSA surface.

## 3.6. Molecular docking

Theoretical studies on the docking of NAZ on the HSA surface were executed to verify our experimental results. To develop the docking strategy for NAZ on the HSA, various protocols were evaluated to

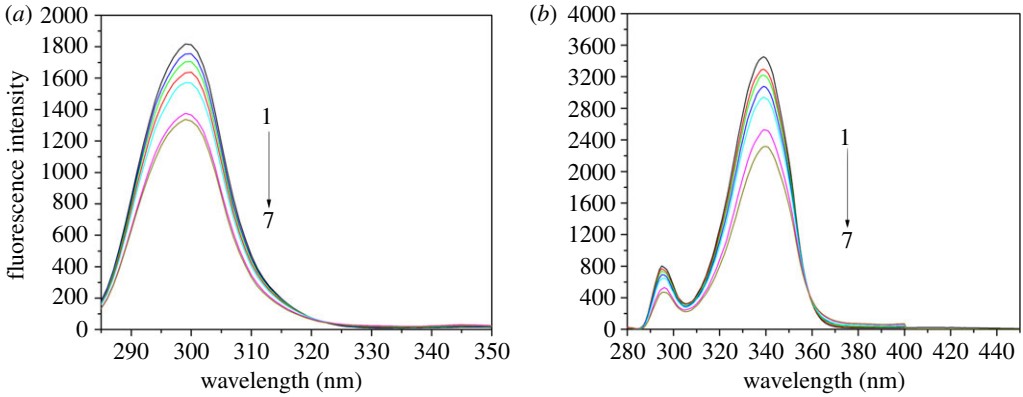

**Figure 6.** The recorded synchronous response of HSA (1.5 µM) at (*a*) $\Delta\lambda = 15$ nm and at (*b*) $\Delta\lambda = 60$ nm, upon addition of NAZ (numbers 1–7 correspond to 0–22.0 µM NAZ concentrations).

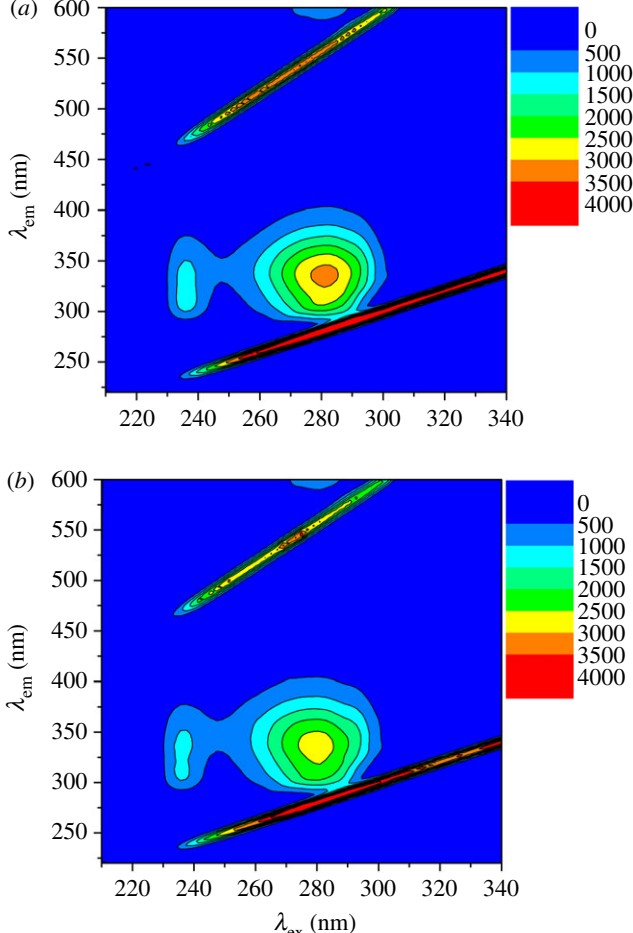

**Figure 7.** Three-dimensional plots of HSA fluorescence (1.5 µM) in the (*a*) absence and (*b*) presence of NAZ (5.5 µM).

achieve the best correlation with experimental data. Two major challenges were faced during the optimization of the docking protocol for NAZ on HSA surface, viz. the binding sites are believed to be conformationally flexible, and most of the available crystal structures have relatively poor resolution. Consequently, the treatment of receptor flexibility in the docking protocol was our major focus, hence residues at the active site were kept flexible (induced fit approach) [38]. Additionally, since PHB and IBP were employed as site markers in the experimental procedure (§3.5), crystal structures of HSA complexed with PHB (PDB ID: 2BXC) and with IBP (PDB ID:2BXG) were defined

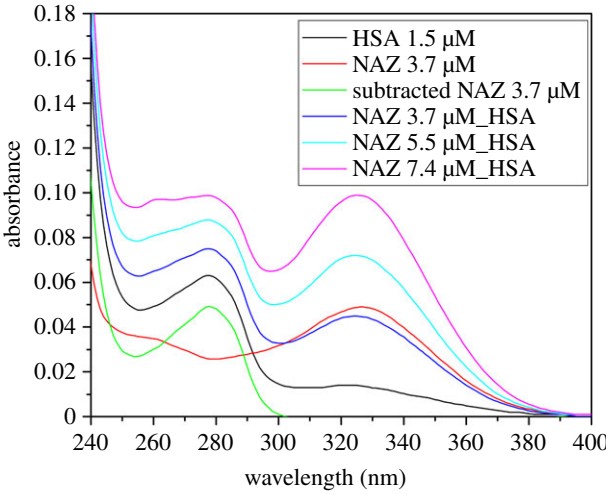

**Figure 8.** UV–vis spectra of NAZ, HSA and the formed complex with the normalized/corrected HSA-NAZ spectrum (subtracted NAZ, 3.7 µM).

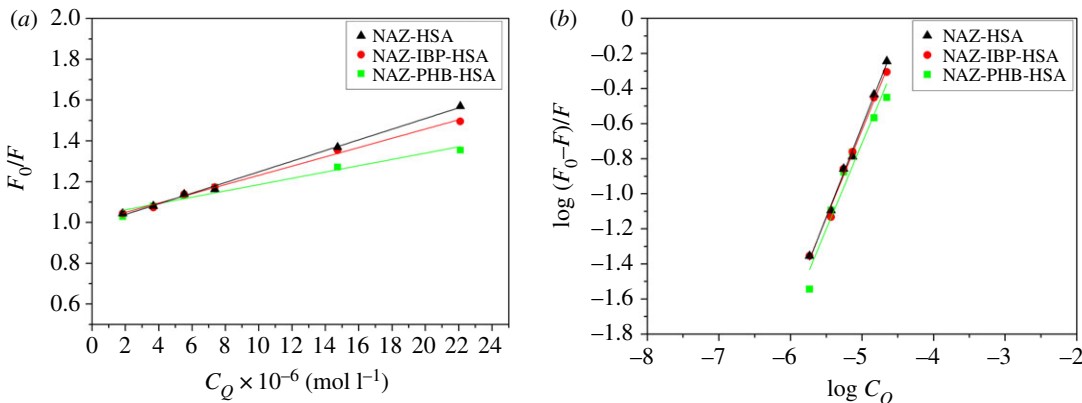

**Figure 9.** (a) Stern–Volmer and (b) double-log graphs for HSA-NAZ binding in the presence and absence of PHB and IBP.

**Table 3.** Main parameters obtained from three-dimensional fluorescence measurements.

| | HSA | | HSA-NAZ | |
|---|---|---|---|---|
| | 1st peak | 2nd peak | 1st peak | 2nd peak |
| peak position ($\lambda_{ex}/\lambda_{em}$) (nm) | 234/336 | 280/336 | 234/336 | 280/336 |
| relative intensity ($IF$) (% reduction from native protein) | 1266.07 — | 3195.95 — | 1138.87 (−10.05%) | 2766.54 (−13.44%) |
| $\Delta\lambda$ (nm) | 102 | 56 | 102 | 56 |

as the total receptor by exclusively selecting the protein part for the 'Define Receptor' function in the MOE® software. In 2BXC crystal, PHB was clustered at the centre of the site I pocket while in 2BXG crystal, IBP was clustered at the centre of the binding pocket of site II and oriented with at least one 'O' atom in the polar patch vicinity. However, IBP also occupied a secondary site at the interface between subdomains IIA and IIB in 2BXG, with this latter site not considered further because the current study only focused on sites I and II as the main binding sites. Selection of the NAZ conformers with the lowest free energy ($\Delta G$) and root-mean-square distance (RMSD) values produced upon binding to HSA was performed based on the London dG and GBVI/WSA dG scoring criteria

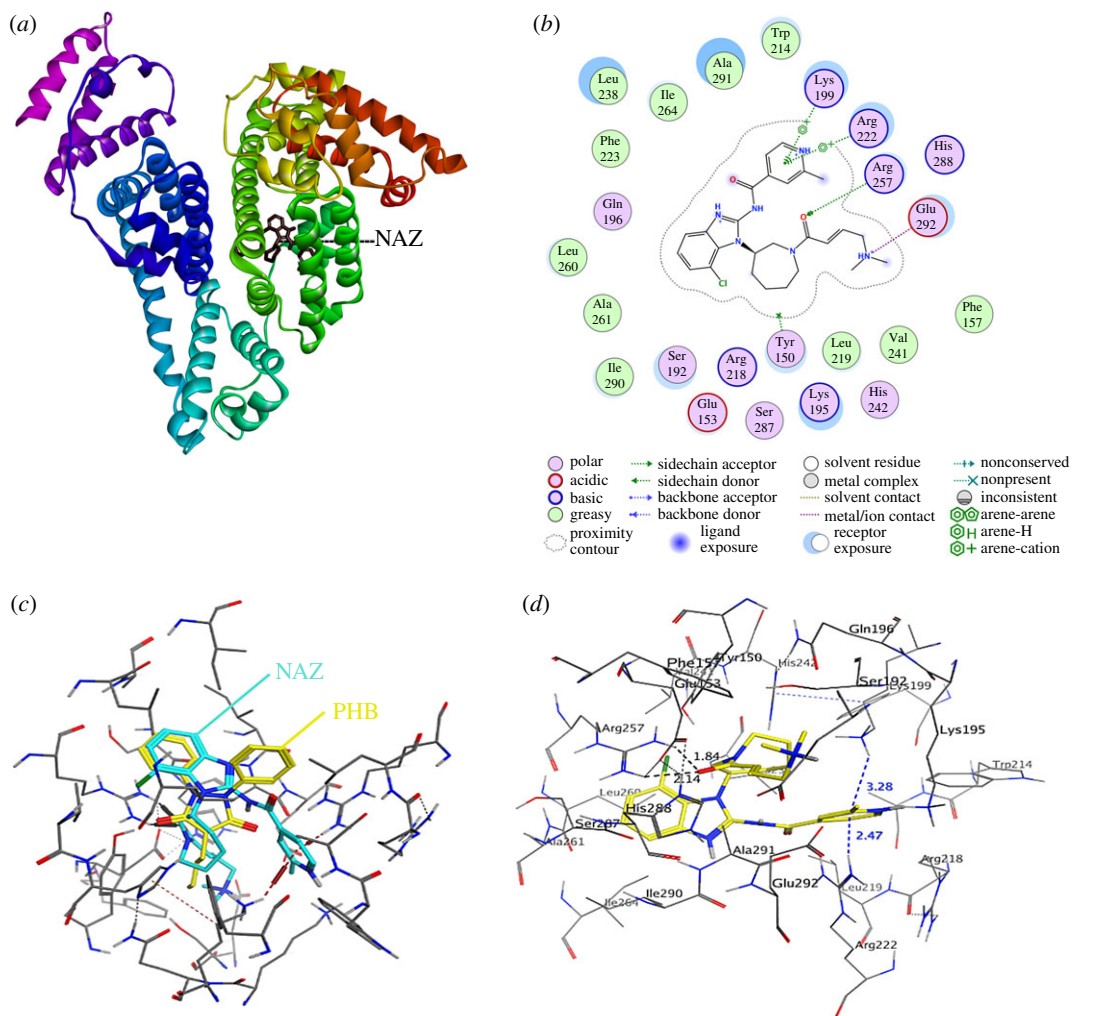

**Figure 10.** Optimum NAZ bound conformer into site I on the HSA surface.

**Table 4.** Main HSA-NAZ binding variables with and without PHB and IBP.

| systems | Stern−Volmer parameters | | double-log formula derived parameters | |
|---|---|---|---|---|
| | $K_{SV} \times 10^4$ (M$^{-1}$) | $r^2$ | $K \times 10^4$ (M$^{-1}$) | $r^2$ |
| HSA-NAZ | $2.71 \pm 0.08$ | 0.9982 | $2.81 \pm 0.07$ | 0.9979 |
| HSA-NAZ + IBP | $2.37 \pm 0.07$ | 0.9979 | $2.41 \pm 0.05$ | 0.9962 |
| HSA-NAZ + PHB | $1.54 \pm 0.14$ | 0.9827 | $1.57 \pm 0.17$ | 0.9776 |

within MOE® modelling software. The binding energies for NAZ on the 2BXC and 2BXG were −25.59 and −6.42 kJ M$^{-1}$, respectively. To further validate the docking procedure, docking of NAZ with two other HSA crystal structures, namely 2BXD (HSA complexed with warfarin; site I marker) and 2BXH (HSA complexed with indoxyl sulfate; site II marker) were performed. The results of NAZ docking to those two crystal structures revealed free energy values of −24.99 and 11.82 kJ M$^{-1}$ for PDB ID: 2BXD and 2BXH, respectively. The defined amino acid residues in the HSA binding pocket for NAZ with the crystal structure of PDB ID: 2BXC are shown in figure 10 and listed in table 5. These results are in agreement with the solution-based investigational data, thereby confirming that NAZ binds to Sudlow site I of HSA and that this interaction is spontaneously driven by electrostatic forces and hydrogen bonding.

**Table 5.** Theoretically determined docking features of HSA-NAZ binding.

| amino acid residues | interaction forces | distance (Å) | aggregate binding energy (kJ mol$^{-1}$) |
|---|---|---|---|
| ARG 257 | H-acceptor | 2.51 | −25.59 |
| ARG 222 | H-acceptor | 3.33 | |
| LYS 199 | pi-H | 3.43 | |
| GLU 292 | H-donor | 2.74 | |

## 4. Conclusion

Insights into the molecular features of the interaction between NAZ and HSA were gained by means of fluorescence and UV–vis spectroscopic tools complemented by molecular docking studies. Fluorescence observations revealed that the interaction of HSA with NAZ induced quenching of the inherent fluorescence of HSA. Mathematical manipulations of the results revealed that this quenching was due to the static formation of an HSA-NAZ complex with a binding constant in the order of $10^4\,M^{-1}$. Moreover, synchronous and three-dimensional fluorescence as well as UV–vis observations of the HSA-NAZ interaction verified the complex formation with no suggested conformational changes in the HSA structure following its binding to NAZ. Competitive fluorescence studies using site markers in addition to molecular docking studies showed that NAZ binds the Sudlow site I on the HSA surface through hydrogen bonding and electrostatic forces. The findings of this study can be of a significant benefit to the continued clinical assessment of NAZ to further understand its pharmacokinetic features.

Data accessibility. The data supporting the results in this article can be accessed from the Dryad Digital Repository: https://dx.doi.org/10.5061/dryad.05c6v93 [39].

Authors' contributions. E.S.G.H. and A.S.A. participated in research designing and experimental work supervision. A.A.A., H.A., A.H.B and R.N.H. performed the optimization, measurements and method validation studies. E.S.G.H. and A.S.A. performed the data analyses. A.S.A., A.A.A. and H.A. prepared the figures. A.H.B. performed the docking studies. E.S.G.H. and A.S.A. wrote the first draft of the manuscript. All authors revised and approved the final form of the manuscript.

Competing interests. The authors declare no competing interests.

Funding. The authors received funding from the Deanship of Scientific Research at King Saud University through Research Group Project no. RG-1435-025.

Acknowledgements. The authors extend their sincere appreciation to the Deanship of Scientific Research at King Saud University for its funding this Research Group no. RG-1435-025.

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
