## [Reviewer comments · Royal Society Open Science]

Review History

RSOS-190922.R0 (Original submission)

Review form: Reviewer 1

Is the manuscript scientifically sound in its present form?

No

Are the interpretations and conclusions justified by the results?

No

Is the language acceptable?

No

Do you have any ethical concerns with this paper?

No

Have you any concerns about statistical analyses in this paper?

Yes

Recommendation?

Major revision is needed (please make suggestions in comments)

Comments to the Author(s)

The authors carried out an interesting work characterizing the binding mode of NAZ with HAS by experimental and theoretical methods. The authors show a great amount of information in the spectroscopic analysis. Nevertheless, the docking part it is short. For example, they never talk about why they chose that HAS crystal, and never talk about the validation of their docking procedure. They need to justify why that crystal?, because of its low X-ray resolution, and why that references ligand?. Only to mention some details.

Review form: Reviewer 2**Is the manuscript scientifically sound in its present form?**

Yes

Are the interpretations and conclusions justified by the results?

Yes

Is the language acceptable?

Yes

Do you have any ethical concerns with this paper?

No

Have you any concerns about statistical analyses in this paper?

No

Recommendation?

Accept with minor revision (please list in comments)

Comments to the Author(s)

In this article, the authors investigated the interactions between Nazartinib (NAZ) a new anti-cancer EGFR tyrosine kinase inhibitor and human serum albumin (HSA) using experimental and theoretical approaches. Experiments of fluorescence quenching and UV-vis spectroscopic analysis, revealed the formation of a statically complex between NAZ and HSA and a binding constant for HSA-NAZ complex was computed in the studied temperature range. From these results, the thermodynamic parameters of the HSA-NAZ interaction were established by linear regression methods, which led to conclude that the non-covalent interaction was of electrostatic-driven type. The authors then identified the specific binding site of NAZ into HSA by ligand displacement studies using fluorescence techniques. They could conclude that NAZ binds into the HSA phenylbutazone site (namely Sudlow site 1). Finally, these experimental results were completed by molecular docking studies, which showed that NAZ binds the Sudlow site I on the HSA surface, through hydrogen bonding and electrostatic forces and not on the Ibuprophen site II.

This work is of some interest and I am favorable for a publication in Royal Society Open Science, provided that the authors provide additional data concerning the docking study.

Since experimental data led to the conclusion that NAZ does not bind into the Sudlow site II of Ibuprofen, the authors could complement their work by the same molecular docking analysis performed on the ibuprofen site. It will thus be possible to compare the docking results found with both sites and get a better idea of the value of the computed aggregate binding energy.

In addition, I found 2 corrections to make into the references:

- Ref. 28: Journal of Molecular LIQUIDS and not Journal of Molecular LIQUIDS;
- Ref 38 should be completed; the doi:10.5061/dryad.05c6v93 is not valid.

Review form: Reviewer 3

Is the manuscript scientifically sound in its present form?

No

Are the interpretations and conclusions justified by the results?

No

Is the language acceptable?

Yes

Do you have any ethical concerns with this paper?

No

Have you any concerns about statistical analyses in this paper?

No

Recommendation?

Reject

Comments to the Author(s)

Please see attached document (Appendix A).

Decision letter (RSOS-190922.R0)

01-Jul-2019

Dear Dr Hassan:

Manuscript ID: RSOS-190922

Title: "Spectroscopic and molecular docking studies reveal binding characteristics of nazartinib (EGF816) to human serum albumin"

Thank you for submitting the above manuscript to Royal Society Open Science. Your paper was sent to reviewers and their comments are included at the bottom of this letter.

In view of the concerns raised by the reviewers, the manuscript has been rejected in its current

form. However, a new manuscript may be submitted which takes into consideration these comments.

Please note that resubmitting your manuscript does not guarantee eventual acceptance, and that your resubmission will be subject to peer review before a decision is made.

Your resubmitted manuscript should be submitted by 29-Dec-2019. If you are unable to submit by this date please contact the Editorial Office.

On behalf of the Subject Editor Professor Anthony Stace and the Associate Editor Mr Andrew Dunn

REVIEWER(S) REPORTS:

Associate Editor Comments to Author ():

RSC Associate Editor:

Comments to the Author:

Although the reviewers make some positive comments, significant improvements are required to the docking studies before this manuscript could be suitable for publication. We would be happy to consider a revised version of this manuscript.

RSC Subject Editor:

Comments to the Author:

(There are no comments.)

Reviewers' Comments to Author:

Reviewer: 1

Comments to the Author(s)

The authors carried out an interesting work characterizing the binding mode of NAZ with HAS by experimental and theoretical methods. The authors show a great amount of information in the spectroscopic analysis. Nevertheless, the docking part it is short. For example, they never talk

about why they chose that HAS crystal, and never talk about the validation of their docking procedure. They need to justify why that crystal?, because of its low X-ray resolution, and why that references ligand?. Only to mention some details.

Reviewer: 2

Comments to the Author(s)

In this article, the authors investigated the interactions between Nazartinib (NAZ) a new anti-cancer EGFR tyrosine kinase inhibitor and human serum albumin (HSA) using experimental and theoretical approaches. Experiments of fluorescence quenching and UV-vis spectroscopic analysis, revealed the formation of a statically complex between NAZ and HSA and a binding constant for HSA-NAZ complex was computed in the studied temperature range. From these results, the thermodynamic parameters of the HSA-NAZ interaction were established by linear regression methods, which led to conclude that the non-covalent interaction was of electrostatic-driven type. The authors then identified the specific binding site of NAZ into HSA by ligand displacement studies using fluorescence techniques. They could conclude that NAZ binds into the HSA phenylbutazone site (namely Sudlow site 1). Finally, these experimental results were completed by molecular docking studies, which showed that NAZ binds the Sudlow site I on the HSA surface, through hydrogen bonding and electrostatic forces and not on the Ibuprophen site II.

This work is of some interest and I am favorable for a publication in Royal Society Open Science, provided that the authors provide additional data concerning the docking study. Since experimental data led to the conclusion that NAZ does not bind into the Sudlow site II of Ibuprophen, the authors could complement they work by the same molecular docking analysis performed on the ibuprofen site. It will thus be possible to compare the docking results found with both sites and get a better idea of the value of the computed aggregate binding energy.

In addition, I found 2 corrections to make into the references:

- Ref. 28: Journal of Molecular LIQUIDS and not Journal of Molecular LIQUIDS;
- Ref 38 should be completed; the doi:10.5061/dryad.05c6v93 is not valid.

Reviewer: 3

Comments to the Author(s)

Please see attached document.

Author's Response to Decision Letter for (RSOS-190922.R0)

See Appendix B.

RSOS-191595.R0

Review form: Reviewer 1

Is the manuscript scientifically sound in its present form?

Yes

Are the interpretations and conclusions justified by the results?

Yes

Is the language acceptable?

Yes

Do you have any ethical concerns with this paper?

Yes

Have you any concerns about statistical analyses in this paper?

No

Recommendation?

Accept as is

Comments to the Author(s)

The work can be accepted for its publication.

Review form: Reviewer 2

Is the manuscript scientifically sound in its present form?

Yes

Are the interpretations and conclusions justified by the results?

Yes

Is the language acceptable?

Yes

Do you have any ethical concerns with this paper?

No

Have you any concerns about statistical analyses in this paper?

No

Recommendation?

Accept as is

Comments to the Author(s)

The authors generally met my expectations: they completed the part relating to docking and they thus performed the docking of NAZ on the second binding site of the HSA protein. Furthermore, the docking results are in agreement with the experimental results.

However, I note that reference 38 (which became reference 39), has not been corrected. The doi "10.5061/dryad.05c6v93" does not exist. If the corresponding link is not accessible, then this reference must be modified or removed.

Decision letter (RSOS-191595.R0)

28-Oct-2019

Dear Dr Hassan:

Title: Spectroscopic and molecular docking studies reveal binding characteristics of nazartinib (EGF816) to human serum albumin

Manuscript ID: RSOS-191595

It is a pleasure to accept your manuscript in its current form for publication in Royal Society Open Science. The chemistry content of Royal Society Open Science is published in collaboration with the Royal Society of Chemistry.

RSC Associate Editor
Comments to the Author:
Please correct Reference 39 before final publication.

Reviewer(s)' Comments to Author:
Reviewer: 1

Comments to the Author(s)
The work can be accepted for its publication.

Reviewer: 2

Comments to the Author(s)

The authors generally met my expectations: they completed the part relating to docking and they thus performed the docking of NAZ on the second binding site of the HSA protein. Furthermore, the docking results are in agreement with the experimental results.

However, I note that reference 38 (which became reference 39), has not been corrected. The doi "10.5061/dryad.05c6v93" does not exist. If the corresponding link is not accessible, then this reference must be modified or removed.

Appendix A

Ref: RSOS-190992

Title: *Spectroscopic and molecular docking studies reveal binding characteristics of nazartinib (EGF816) to human serum albumin*

In this manuscript, the authors present an experimental study characterizing the binding of the drug NAZ to the protein HSA. The authors use computational molecular docking to gain insight into a plausible binding site for NAZ. Initial characterization involves fluorescence quenching which suggests that the ligand is either forming a static non-fluorescent complex or is diffusing through the system. They then use stern-volment, LB and double log correlations to gain insight into the mechanism and establish that the HAS-NAZ complex does indeed form. The thermodynamics of binding is then evaluated by observing changes to the binding constants with changes in temperature.

While the manuscript is mostly organized well, with the methods being presented logically and the results and interpretations discussed in detail, the manuscript requires significant improvements in order to be considered for publication.

However, there are some general issues that need to be addressed before final publication of the manuscript:

1. Figures. The insets in Figure 2 is not legible at all. The font size for the axes are too small to be legible.
2. Molecular docking: Can the authors clarify whether or not the residues were maintained to be flexible during the docking process? The MOE software should enable this with the induced fit docking protocol. Can the authors please highlight the important residues lining the binding site in Figure 10c? The following publications might be useful in guiding the authors to improving the docking protocol and results in the manuscript.
 1. Sambasivarao, S. V. *et al.* Acetylcholine Promotes Binding of α -Conotoxin MII at $\alpha 3\beta 2$ Nicotinic Acetylcholine Receptors. *ChemBioChem* **15**, 413–424 (2014).
 2. Bharadwaj, V. S., Dean, A. M. & Maupin, C. M. Insights into the Glycyl Radical Enzyme Active Site of Benzylsuccinate Synthase: A Computational Study. *J. Am. Chem. Soc.* **135**, 12279–12288 (2013).
 3. Bharadwaj, V. S., Kim, S., Guarnieri, M. T. & Crowley, M. F. Different Behaviors of a Substrate in P450 Decarboxylase and Hydroxylase Reveal Reactivity-Enabling Actors. *Sci. Rep.* **8**, 12826 (2018).
 4. C. Schutt, T., S. Bharadwaj, V., M. Granum, D. & Mark Maupin, C. The impact of active site protonation on substrate ring conformation in *Melanocarpus albomyces* cellobiohydrolase Cel7B. *Phys. Chem. Chem. Phys.* **17**, 16947–16958 (2015).

Appendix B

Dear Editor

I am deeply appreciating your efforts in considering our work. Kindly find below our point-by-point responses to those comments raised by the reviewers. All changes are highlighted in the revised manuscript

Looking forward to hearing from you.

Sincerely,

Eman S. G. Hassan, Ph.D

Associate Editor Comments to Author ():

RSC Associate Editor:

Comments to the Author:

Although the reviewers make some positive comments, significant improvements are required to the docking studies before this manuscript could be suitable for publication. We would be happy to consider a revised version of this manuscript.

Response:

Thank you for considering our revised version of the manuscript, now the docking study has been improved and the details are now added to the revised manuscript

Reviewer: 1

The authors carried out an interesting work characterizing the binding mode of NAZ with HAS by experimental and theoretical methods. The authors show a great amount of information in the spectroscopic analysis. Nevertheless, the docking part it is short.

For example,

(1) they never talk about why they chose that HAS crystal

Response (1):

- PHB and IBP were chosen as probes in section 3.3.1 of this study, and the crystals 2BXC (HSA complexed with PHB)/2BXG (HSA complexed with IBP) were defined as the total receptor by exclusively selecting the protein part for the “Define Receptor” function. In 2BXC crystal, PHB was clustered at the center of the site I pocket. In 2BXG crystal, IBP was clustered at the center of the binding pocket of site II and oriented with at least one “O” atom in the polar patch vicinity. However, IBP also occupied a secondary site at the interface between subdomains IIA and IIB in 2BXG. The second site was not considered further because the current study only focused on sites I and II as the main binding sites. The binding energies for NAZ on the 2BXC and 2BXG were -25.59 kJ M^{-1} and 11.82 kJ M^{-1} , respectively.

(2) and never talk about the validation of their docking procedure.

Response (2):

- To further validate our docking procedure, we also performed the docking with two more crystal structures, namely 2BXD (HSA complexed with warfarin; site I marker) and 2BXH (HSA complexed with indoxyl sulfate) and analyzed the binding modes and binding energies. Free energies obtained from docking studies with PDB ID: 2BXD were shown to be -24.99 kJ M^{-1} for NAZ. Also, studies based on PDB ID: 2BXH showed 11.82 kJ M^{-1} for NAZ, when they bound to site I and site II, respectively (data not shown). Thus, the same binding modes are obtained even if we tried with other PDB entries of HSA since they have similar binding pockets for all the crystal structures. Further, molecular displacement studies confirm that NAZ binds site I (Fig. 5), in agreement with the computationally determined docking studies. Thus the results of molecular modeling study are in good agreement with the results of the experimental study. Unfortunately, currently we lack the expertise and resources to run Molecular Dynamics studies to further confirm the acquired results.

(3) They need to justify why that crystal?, because of its low X-ray resolution, and why that references ligand?. Only to mention some details.

Response (3):

- To develop the docking strategy for NAZ on the HSA, we began by evaluating a variety of protocols to find the best correlation with experimental data. The major challenges are (1) the binding sites are believed to be conformationally flexible, and (2) most of the available structures have relatively poor resolution. For these reasons, the treatment of receptor flexibility in the docking protocol was our major focus.

Reviewer: 2

In this article, the authors investigated the interactions between Nazartinib (NAZ) a new anti-cancer EGFR tyrosine kinase inhibitor and human serum albumin (HSA) using experimental and theoretical approaches. Experiments of fluorescence quenching and UV-vis spectroscopic analysis, revealed the formation of a statically complex between NAZ and HSA and a binding constant for HSA-NAZ complex was computed in the studied temperature range. From these results, the thermodynamic parameters of the HSA-NAZ interaction were established by linear regression methods, which led to conclude that the non-covalent interaction was of electrostatic-driven type. The authors then identified the specific binding site of NAZ into HSA by ligand displacement studies using fluorescence techniques. They could conclude that NAZ binds into the HSA phenylbutazone site (namely Sudlow site 1). Finally, these experimental results were completed by molecular docking studies, which showed that NAZ binds the Sudlow site I on the HSA surface, through hydrogen bonding and electrostatic forces and not on the Ibuprofen site II.

This work is of some interest and I am favorable for a publication in Royal Society Open Science, provided that the authors provide additional data concerning the docking study.

1. Since experimental data led to the conclusion that NAZ does not bind into the Sudlow site II of Ibuprofen, the authors could complement they work by the same molecular docking analysis

performed on the ibuprofen site. It will thus be possible to compare the docking results found with both sites and get a better idea of the value of the computed aggregate binding energy.

Response (1):

- As per the reviewer's suggestion, docking study was also performed using the crystal 2BXG (HSA complexed with IBP) and the results are now added in the context of the revised manuscript

2- In addition, I found 2 corrections to make into the references:

- Ref. 28: Journal of Molecular LIQUIDS and not Journal of Molecular LIQUIDS;
- Ref 38 should be completed; the doi:10.5061/dryad.05c6v93 is not valid.

Response (2):

- Corrections are now made in the revised manuscript as per the reviewer's suggestion

Reviewer 3:

In this manuscript, the authors present an experimental study characterizing the binding of the drug NAZ to the protein HSA. The authors use computational molecular docking to gain insight into a plausible binding site for NAZ. Initial characterization involves fluorescence quenching which suggests that the ligand is either forming a static nonfluorescent complex or is diffusing through the system. They then use stern-volmer, LB and double log correlations to gain insight into the mechanism and establish that the HASNAZ complex does indeed form. The thermodynamics of binding is then evaluated by observing changes to the binding constants with changes in temperature. While the manuscript is mostly organized well, with the methods being presented logically and the results and interpretations discussed in detail, the manuscript requires significant improvements in order to be considered for publication. However, there are some general issues that need to be addressed before final publication of the manuscript:

1. Figures. The insets in Figure 2 is not legible at all. The font size for the axes are too small to be legible.

Response (1):

- Figure 2 is now edited in the revised manuscript for better clarification as per the reviewer's suggestion

2. Molecular docking: Can the authors clarify whether or not the residues were maintained to be flexible during the docking process? The MOE software should enable this with the induced fit docking protocol. Can the authors please highlight the important residues lining the binding site in Figure 10c?

Response (2):

- Study details are now clarified in the text of the revised manuscript, Figure 10d has now been added to the revised manuscript which includes the important residues involved in the NAZ-HSA binding, as figure 10c shows the super imposed configurations of NAZ and PHB in the HSA

binding pocket, hence it would be unclear if labeled the AA residues in such figure, so we added a new one.

3. The following publications might be useful in guiding the authors to improving the docking protocol and results in the manuscript.

1. Sambasivarao, S. V. et al. Acetylcholine Promotes Binding of α -Conotoxin MII at $\alpha 3\beta 2$ Nicotinic Acetylcholine Receptors. *ChemBioChem* 15, 413–424 (2014).

2. Bharadwaj, V. S., Dean, A. M. & Maupin, C. M. Insights into the Glycyl Radical Enzyme Active Site of Benzylsuccinate Synthase: A Computational Study. *J. Am. Chem. Soc.* 135, 12279–12288 (2013).

3. Bharadwaj, V. S., Kim, S., Guarnieri, M. T. & Crowley, M. F. Different Behaviors of a Substrate in P450 Decarboxylase and Hydroxylase Reveal Reactivity Enabling Actors. *Sci. Rep.* 8, 12826 (2018).

4. C. Schutt, T., S. Bharadwaj, V., M. Granum, D. & Mark Maupin, C. The impact of active site protonation on substrate ring conformation in *Melanocarpus albomyces* cellobiohydrolase Cel7B. *Phys. Chem. Chem. Phys.* 17, 16947–16958 (2015).

Response (3):

- We appreciate the reviewer's suggestion of using such valuable references to enrich the discussion, and the used references are now cited in the text of the revised manuscript